# Preparation and Evaluation of Vitamin D3 Supplementation as Transdermal Film-Forming Solution

**DOI:** 10.3390/pharmaceutics15010039

**Published:** 2022-12-22

**Authors:** Majd Kittaneh, Moammal Qurt, Numan Malkieh, Hani Naseef, Ramzi Muqedi

**Affiliations:** 1Faculty of Pharmacy, Nursing and Health Professions, Birzeit University, Ramallah P.O. Box 3570, Palestine; 2Jerusalem Pharmaceuticals, Ramallah P.O. Box 71939, Palestine

**Keywords:** vitamin D3, transdermal, percutaneous, film-forming solution, drug release, drug permeation, cholecalciferol, drug delivery

## Abstract

Vitamin D3 is available in oral and injectable dosage forms. Interest in the transdermal route as an alternative to the oral and parenteral routes has grown recently. In this study, several film-forming solutions for the transdermal delivery of vitamin D3 were prepared. They contained 6000 IU/mL of vitamin D3 that formed a dry and acceptable film in less than 5 min after application. The formulations consisted of ethanol and acetone 80:20, and one or more of the following ingredients: Eudragit L100-55, PVP, PG, limonene, oleic acid, camphor, and menthol. Vitamin D3 release was studied from both the film-forming solution and pre-dried films using a Franz diffusion cell. The film-forming solution released a significant amount of vitamin D3 compared to the dry film, which is attributed mostly to the saturation driving force due to the evaporation of volatile solvents. In vitro permeation studies through artificial skin Strat M^®^ membrane revealed that the cumulative amount of vitamin D3 permeated after 24 h under the experimental conditions was around 800 IU across 3.14 cm^2^. The cumulative permeation curve showed faster permeation in earlier stages. Young’s modulus, viscosity, and pH of the formulations were determined. Most of the formulations were stable for 3 weeks.

## 1. Introduction

Over time, interest in the transdermal route (TDR) as an alternative to the oral and parenteral routes has grown significantly due to the advantages offered. TDR can avoid first-pass metabolism, overcome low oral bioavailability, provide long-term controlled drug release, avoid gastrointestinal side effects, and it has more predictable drug delivery. Comparing it with the parenteral route, it is considered more convenient because it is self-administered, non-invasive, and has a lower risk of disease transmission; moreover, the termination of the dose is simple and can be easily achieved by removing the transdermal system [1,2,3].

The skin has a unique structure composed of three main layers: the epidermis, the dermis, and the hypodermis. The outer layer of the epidermis is the stratum corneum (SC). The diffusion through this metabolically inactive dead layer is the rate-limiting step. The intercellular lipid in the SC is the essential pathway for percutaneous absorption [4]. 

The synthetic artificial membrane Strat-M^®^ is manufactured to mimic human skin. It is considered a substitute for human and animal skin in permeation studies [5,6,7,8]. Strat-M^®^ consists of several layers of polyester sulfone that differ in diffusivity. The outermost layer is a tightly packed surface layer which resembles the function of the SC. Underneath this layer, there are two layers of polyethersulfone that resemble the dermis. At the bottom, there is a more diffusive polyolefin layer that resamples the subcutaneous fat layer. The porous structure of this membrane provides a permeability gradient to mimic the permeability of human skin [5,9].

Strat-M^®^ has been used in several publications [10,11,12,13,14]. It was used to predict the permeation of both lipophilic and hydrophilic drugs; the permeability of the hydrophilic compound was higher. These studies showed that the permeability of Strat-M^®^ and human skin is comparable. This makes Strat-M^®^ a reasonable alternative to predict drug permeation through real human skin [5].

Vitamin D3 is considered a hormonal steroid; naturally, it is synthesized upon skin exposure to ultraviolet B (UVB) radiation, where 7-hydrocholesterol converts into cholecalciferol (vitamin D3) [15]. It has an essential role in bone mineralization and skeleton growth, and its deficiency is associated with several illnesses, such as type 1 diabetes mellitus, rheumatoid arthritis, Crohn’s disease, several lung diseases, and others [16,17,18,19].

Due to current lifestyle habits, nutritional supplementation has become the main source of vitamin D3; it is available in parenteral and oral dosage forms, and the recommended daily allowance (RDA) is 400 IU/day (or 10 µg/day) [16]. Vitamin D3 possesses some properties that make it a suitable candidate for transdermal delivery. It has a low melting point ≈ 83–86 °C [20,21], it is considered very potent (RDA is small) [22], and it has a relatively low molecular weight of 384.64 g/mol. It is insoluble in water but very soluble in several other solvents [16]. Vitamin D3 also does not produce any form of inflammation [23]; however, it is a very lipophilic substance (log P = 10.2) [16], which presents a challenge that could be overcome with chemical penetration enhancers (PEs) or any other formulation approaches.

Several recent studies were conducted to investigate the transdermal delivery of vitamin D3 through various strategies. D’Angelo Costa GM et al. used several chemical penetration enhancers (PEs) (propylene glycol, ethoxydiglycol, isopropyl palmitate, alcohol, and soybean lecithin) in cream and gel formulation. Even with the usage of combined PEs, vitamin D3 in the cream-based formulation remained at the surface. On the other hand, the retention of vitamin D3 in the skin layers was significant in the gel-based formulation. They concluded that vitamin D3 retention was due to its high lipophilic properties, which may be helpful for psoriasis, and more potent PEs or more hydrophilic analogs should be investigated to enhance transdermal penetration [24]. In another study, Ahmed Alsaqr and co-workers studied the penetration of vitamin D3 from ointment preparation that contained oleic acid (OA) or dodecylamine as PEs. OA showed no significant improvement in penetration compared with the control. However, the usage of dodecylamine improved the transdermal penetration of vitamin D3, and especially after pretreatment of the skin with 50% ethanol. The synergistic effect of dodecylamine and ethanol resulted in penetration of 760 ng (30.4 IU) of vitamin D3 as an accumulative amount. The RDA of vitamin D3 could be achieved when covering the skin with 3.6 cm^2^ of the formulation [16]. In another study, vitamin D3 was prepared as a reservoir-type transdermal adhesive patch of 40 cm^2^ size, it contained transcutol (diethylene glycol monoethyl ether) as a PE. The patch delivered more than 2000 µg within 5 h, and more than 20,000 µg within 24 h through unbroken intact living skin [25]. Other researchers used the polymeric nanoparticle (TyroSphere) and coated microneedles for transdermal penetration of vitamin D3. Other studies used more hydrophilic analogs of vitamin D3 (Calcitriol, 25-hydroxy D3, and Oxacalcitriol), which also showed promising results [26].

In general, the transdermal film-forming solution (FFS) is composed of the solute, polymers, and other components dissolved in a highly volatile solvent. After application of the FFS on the skin, the volatile solvent evaporates, leaving a thin residual transparent film containing the solute and other ingredients (such as the film-forming polymers (FFPs), plasticizers, non-volatile solvents, and PEs). During evaporation, the concentration of the solute increases towards saturation or even super saturation, and this can result in an increase in the permeation rate. FFS can also provide an invisible depot for solutes/drugs that is expected to permit sustained release for a long time. It is an attractive transdermal dosage form with better patient compliance and offers a decreased risk of drug/clothes contamination when compared to other transdermal preparations such as creams and ointments [27,28].

The aim of the present study was to formulate and evaluate an FFS which is able to deliver vitamin D3 supplementation, and after rapid solvent evaporation, a thin, almost transparent, adherent, non-sticky, and flexible film can be obtained.

## 2. Materials and Methods

### 2.1. Materials

Cholecalciferol (vitamin D3) was received as a gift sample from Jerusalem Pharmaceuticals Co. Ltd. (Ramallah, Palestine). Eudragits (L100-55, L100, and S100) were donated by Evonik. Polyvinylpyrrolidone (PVP) k30, polyethylene glycol (PEG) 400, propylene glycol (PG), sodium lauryl sulfate (SLS), camphor, and menthol, all pharmaceutical grade, were donated by Jerusalem Pharmaceuticals Co. Ltd (Ramallah, Palestine). Oleic acid (OA), isopropyl myristate (IPM), limonene, eucalyptol and diethylene glycol monoethyl ether, ethanol 99.9%, ethyl acetate (EA), isopropyl alcohol (IPA), acetone disodium hydrogen phosphate, and potassium dihydrogen phosphate were purchased from Sigma-Aldrich, Glasgow, UK. Polyamide membranes SUPELCO^®^ (0.45 µm) were purchased from Sigma-Aldrich, Glasgow, UK. Strat M^®^ artificial membranes were purchased from Merck Millipore, Merck Life Science UK Limited (Gillingham, UK).

### 2.2. Instrumentation

A high-performance liquid chromatogram (HPLC) Alginate 1200 system, Santa Clara, CA 95051, USA, was used for sample analysis. It was equipped with a Hypersil Gold Thermo Scientific C18 (250 cm × 4.6 mm) 5 μm column and a UV/VIS detector. A Franz diffusion cell (ORCHID ScientificTM, India) was used for diffusion studies, along with a Cannon-Fenske Ostwald viscometer (Cannon Instrument, State College, PA 16803, USA), climate chamber, (BINDER, Tuttlingen, Germany), analytical balance (Mettler Toledo, Zürich, Switzerland), and refrigerator (Beko, Istanbul, Turkey).

### 2.3. HPLC Analysis Method

Two calibration curves were constructed to cover high and low vitamin D3 concentrations during diffusion studies. Stock solution of vitamin D3 was prepared by dissolving an exact amount of oil containing vitamin D3 in a measured volume of ethanol, and then the mixture was sonicated for 15 min to guarantee complete dissolving. Two sets of standard solutions were prepared by serial dilution. From the R^2^ of the regression line of the calibration curve, we checked the linearity that covers the studied concentration range. The limit of detection (LOD) and the limit of quantification (LOQ) were calculated from the low vitamin D3 concentration curve based on the standard deviation of the intercept and the slope of the calibration curve [29].

#### Chromatographic Conditions

The analysis was carried out on an Alginate series HPLC system. The analytical column was C18, 5 µm with a detection wave length of 265 nm. The operating temperature of the column was set at 25 °C. The mobile phase was acetonitrile 100%. The injection volume was 100 µL and the flow rate was 1.5 mL/min throughout the analysis. The retention time of vitamin D3 was about 9 min.

### 2.4. Formulation

Volatile solvents were selected from the class 3 residual solvent list established in the USP [30] (ethanol, acetone, ethyl acetate, and isopropyl alcohol), and then several primary trial formulations (PTFs) were developed. The film properties of these PTFs were characterized and tested to meet the acceptance criteria in Table 1. Then, the pH, viscosity, and elasticity of the successful trial formulations (STFs) were evaluated. The vitamin D3 was then added to these formulations to obtain a concentration of 6000 IU/mL. These formulations were then carefully evaluated to ensure the preservation of proper film characteristics; afterwards, release and diffusion studies were conducted.

### 2.5. General Method of Preparation

All the components were weighed, added to a volumetric flask containing 50% to 90% of the evaporating solvent, then continually shaken and sonicated (5–60 min, 50/60 Hz, and without heating) to achieve full solvation. The solvent was then added until the volume was reached. At the end, it was sonicated again for a few seconds to ensure complete homogenization. The eutectic mixture was freshly prepared by weighing equal amounts of camphor and menthol. The binary mixture was grounded in a circular motion in a glass mortar and pestle for 5 min until all solid particles melted into one liquid phase.

### 2.6. Preparation of the Primary Trial Formulations (PTFs)

In order to obtain acceptable film properties, several PTFs were prepared using a polymer or a combination of film-forming polymers, a solvent or mixture of solvents, and plasticizers and penetration enhancers, as shown in Table 2. Formulations (X1–X4) were simple polymeric solutions prepared by dissolving 5% of Eudragit L100-55, Eudragit L100, Eudragit S100, or PVP in ethanol as a volatile solvent. To enhance polymeric films’ flexibility and/or adhesiveness, 2% of PEG 400, OA, or PG was dissolved in ethanol, as shown in XP1–XP9. In order to decrease the drying time, a binary evaporating solvent composed of ethanol and acetone (80:20 *w*/*w*) was used as a solvent, as can be seen in the formulation later on. Other ingredients were added to the solvent in an attempt to improve film properties and permeation.

### 2.7. Characterization of PTFs

The PTFs were evaluated to meet the film acceptance criteria shown in Table 1. Polymeric films were prepared to evaluate their drying time, stickiness, adhesiveness, and cosmetic appearance.

#### 2.7.1. Drying Time Stickiness and Flexibility

A total of 100 µL of polymer solution was spread on a slide of glass to cover a predefined area of (5 × 2 cm^2^).

Drying time: Immediately after the application of the liquid preparation, the timer was started. The decrease in weight was monitored on an analytical scale balance. When the loss in the weight was ≤10^−4^ mg/15 s, the drying time was recorded.

Stickiness property: After the film was dried, the stickiness property was evaluated by pulling a metallic ball covered with cotton (weight 7 g, diameter 16 mm) along the film three consecutive times. The film should be non-sticky 5 min after the application. The amount of fiber is proportional to the stickiness property. The film was considered non-sticky only if no fiber was left on the film (Figure 1).

#### 2.7.2. Cosmetic Appearance

The film formed on the slide of glass was considered acceptable if it was clear and transparent/semi-transparent.

#### 2.7.3. Adhesiveness

A polymer film was prepared by pouring a fixed amount of polymer solution into a silicon mold. The mold was kept on a flat surface at room temperature without moving until the film was completely dry. The films that failed to stay in contact with mold were considered to have low adhesiveness. The films that were attached were then removed by tweezers. According to the ease of film removal, the adhesiveness of the film was evaluated as:Low: Not attached/removed without effort.Good: Moderate attachment/removed with some effort.High: Strong attachment/removed with effort.

#### 2.7.4. Flexibility

The film was removed out of the mold and then evaluated for flexibility by being bent, rolled up, and twisted. The film was considered flexible if it could be bent easily and remained intact without breaking or cracking (Figure 2). The film’s flexibility was divided into three categories:Low: Bent, rolled, and twisted with effort and could be broken or cracked easily.Good: Bent, rolled, and twisted with little effort and without breaking.High: Bent, rolled, and twisted easily without effort and without breaking.

#### 2.7.5. pH, Viscosity, and Elasticity

Formulations that fulfilled the parameters were referred to as successful trial formulations and labeled as STFs. These formulation were subjected to further characterization such as pH, viscosity, and elasticity.

pH test: 1 mL from each STF was diluted in 100 mL purified water. The pH of the supernatant was measured by a pH meter at room temperature.

Viscosity test: The Cannon-Fenske Ostwald viscometer was used to measure the viscosity of STFs. The time needed for the solution to pass between the two timing point at 25 ± 1 °C was recorded. The experiment was performed three times (*n* = 3) for each STF, and the average time was recorded (*t*_1_). The test was also performed for water (*n* = 3) to obtain (*t*_2_). Equation (1) was used to calculate the viscosity of the STF (*η*_1_).
(1)$$\frac{\eta_{1}}{\eta_{2}}=\frac{\rho_{1}t_{1}}{\rho_{2}t_{2}}$$
where:
*η_2_*: Viscosity of water;$\rho_{1}$: Density of tested liquid;$\rho_{2}$: Density of water.

Young’s modulus: To determine the modulus of elasticity (Young’s modulus (E)), a thick film was prepared in a mold. A rectangular piece was cut from it. The cross-sectional area (A) was calculated in m^2^. A paperclip was clamped to both ends of the rectangle sheet (Figure 3). The initial length (L○) between the two clamps was measured. Weights were attached to the lower clamp, and the increase in length (∆L) was calculated. The force (F) in Newtons (*n*) was calculated by multiplying the weights in kilograms by the acceleration due to gravity. The weights were removed and the length was re-measured to check whether the film had returned to its initial length or not, and to ensure that the film was remaining in the elastic region with no deformation. We increased the hanging weights gradually and recorded the increase in the length until the increase was significantly greater for the hanging weights. By using Equation (2), and from the stress–strain curve, the value of E was calculated. It was obtained from the slope of the straight line during the elastic region in the stress–strain curve [36].


(2)
$$\mathrm{Young’s}\;\mathrm{modulus}\;(E)=\frac{\mathrm{Stress}\;\left(\sigma \right)}{\mathrm{Strain}\;\left(\varepsilon \right)}$$


For the transdermal thin films, no limits are available yet in the literature, while for transdermal patches available on the market, it ranges between 4 and 501 N/mm^2^ [37].

### 2.8. Vitamin D3 Loading

Vitamin D3 (6000 IU/mL) was loaded in the STFs. An exact amount of oil containing vitamin D3 was first dissolved in the volatile solvent, and then the general method mentioned in Section 2.5 was followed. These formulations were then evaluated to check whether these films will still meet the acceptance criteria or not. Formulations that met the acceptance criteria were further studied for drug release and permeation. A polymeric film containing vitamin D3 and polymer was prepared as a reference in the release study.

### 2.9. Release and Permeation Studies

Regarding the release study, the receptor fluid was a mixture of ethanol and PBS (50:50 *w*/*w*). According to the literature, sink condition was maintained in a mixture of phosphate buffer saline (PBS) (pH 7.4) and ethanol (50:50 *w*/*w*) [24]. The PBS (pH = 7.4) was prepared according to the European Pharmacopeia [38]. The heating jacket of the Franz diffusion cell was set to reach 32 ± 1 °C and the receptor fluid was degassed and heated to reach the same temperature. Then, it was filled in the receptor compartment. The rotation of the magnetic stirrer was adjusted to 600 rpm to provide adequate mixing and keep the sink condition. After being soaked in the receptor fluid for 30 min, the polyamide membranes were mounted on the receptor compartment. The donor compartment was then joined to it after applying a rubber ring to the membrane’s edge. The donor compartment was secured with a metal clamp and tightened. Release studies were performed for both the liquid formulations and for the dry films. Using a volumetric pipette (Hischerman, Germany), 2 mL of the liquid formulations was loaded into the donor compartment. For the dry films, the film was deposited into the donor chamber as a mold before assembling it with the film as one unit. Samples were withdrawn from the sampling port at each timing point during the 6 h of the experiment. The sampling time points for liquid formulations were 1, 2, 3, 4, 5, and 6 h, and for dry films were 0.5 h, 1 h, 2 h, 3 h, 4 h, and 6 h. An exception was made for one of the liquid formulations, where samples were taken at 0.25 h and 0.5 h to study the early release behavior.

The concentration of permeated vitamin D3 in the samples was determined via HPLC, and the cumulative released amount per unit area (Q) was calculated. Based on Equation (3), the diffused amount of drug in the receptor compartment through the effective surface area (S) was calculated, in addition to the previous amount removed from the receptor compartment during sampling. (C_n_) and (C_col_) are the concentration of drug in the receptor compartment and the concentration of drug in the sample, respectively. (V_R_) and (V_i_) are the receptor compartment volume and the sample volume, respectively [39].
(3)$$Q\;=\;\frac{C_{n}V_{R}+\;\sum_{i=1}^{i=n-1}C_{i}V_{\mathrm{col}}}{S\;}$$
where S = 3.14 cm^2^ and V_R_ = 20 mL.

### 2.10. In Vitro Permeation Study Using Strat M^®^ Membrane

The parameters utilized in liquid release studies were also applied in the in vitro permeation study. However, the artificial membrane (Strat M^®^), a non-animal based model for transdermal diffusion, was used instead of the polyamide membrane. Permeation studies were conducted on the liquid primary formulations. A total of 0.5 mL containing 3000 IU was dispensed in the donor compartment. For each trial, the test was performed in triplicate (*n* = 3). Samples were taken at 0.5 h, 1 h, 2 h, 3 h, 4 h, 6 h, 7 h, 8 h, and 24 h. The cumulative amount in the receptor compartment was calculated according to Equation (3).

### 2.11. Stability Studies

The stability of the primary formulations was studied under four storage conditions: long-term (25 ± 2 °C/60% RH ± 5%), intermediate (30 ± 2 °C/65% RH ± 5% RH), accelerated (40 ± 2 °C/75% RH ± 5% RH), and in refrigerator (5 ± 3 °C) at zero time and after incubation for 2 and 3 weeks. Each formulation was filled into individual amber glass vials (type I) protected from light for each time period for each storage condition. The physical appearance was investigated by visual inspection if color modification was observed or precipitates were retained. The amount of vitamin D3 was determined in triplicate by HPLC.

## 3. Results and Discussion

### 3.1. HPLC Analysis

#### Linearity, Range, and Sensitivity

Plotting the concentration against the corresponding average peak areas in Figure 4 and Figure 5 showed the regression line equation for calibration curves. The R^2^ for low and high vitamin D3 calibration curves were very close to 1. This indicates a linear relationship between the concentrations and peak areas over the ranges 0.49–12.05 IU/mL and 12.3–409.4 IU/mL, respectively.

The LOD and LOQ obtained from the low vitamin D3 concentration curves were 0.098 IU/mL and 0.296 IU/mL, respectively. These values are low enough to be relied upon during the release and permeation studies.

### 3.2. Formulation Development

#### 3.2.1. The Primary Trial Formulations (PTFs)

The polymeric trial formulations (X1–X4): Eudragit L100-55 (5%) in X1 and PVP (5%) in X4 formed clear and transparent films. The invisible cosmetic appearance is preferred for patient acceptance and compliance. On the other hand, Eudragit L100 (5%) in X2 and Eudragit S100 (5%) in X3 formed white films. PVP formed a very adhesive film which looked like a glue that could not be removed from the mold; thus, its flexibility could not be determined. All these trial formulations at this primary stage failed to achieve the acceptance criteria, as shown in Table 3.

The initial trial formulations (XP1–XP9): PG (2%) was used in formulations XP3, XP5, and XP8; PEG (2%) in XP1, XP4, and XP7; and OA (2%) in XP2, XP6, and XP9 in an attempt to enhance film flexibility and adhesiveness. Opposite from what was expected, OA (2%) had no effect on flexibility and a low effect on adhesiveness. The white color of the polymeric films Eudragit L100 and Eudragit S100 changed to transparent or opaque. After ethanol evaporation, the film-forming polymers were still dissolved in the non-evaporating solvents (PG, PEG, or OA); this can explain the reason for color changing. The addition of PEG and PG improved the flexibility of Eudragit L100-55 (in XP1 and XP3, respectively) and also slightly improved the flexibility of Eudragit S100 (in XP7 and XP8, respectively). On the other hand, PEG and PG had a negligible effect on the flexibility of Eudragit L100 (in XP4 and XP5, respectively). Although film properties were improved, the drying times for all trial formulations at this stage were still more than the acceptable limit (Table 3).

The modified trial formulation (P1–P9): The use of ethanol and acetone (80:20) in the modified trial formulations decreased the drying time significantly compared with the polymeric and initial trial formulations. This was expected because of the boiling point of acetone, which is less than that for ethanol (56.2 °C vs. 78.15 °C, respectively) [40]. By decreasing the drying time, P1 and P3 met the acceptance criteria. As shown in Table 3, the drying time was less than 5 min, and they formed non-sticky, transparent, adhesive, and flexible films.

The complex trial formulations (A1–A7), (B1–B4), (C1–C4), and (D1–D5): In an attempt to enhance the film properties, the percentage of Eudragit L100-55 was increased to 8% in formulations A1–A6 in the presence of a binary combination of PEG, PG, IPM, and OA (2% each). These trials were prepared to increase the flexibility and adhesiveness of the films but at the same time keep them non-sticky. The results revealed that the combination between PEG and PG increased the chance of stickiness (A3). However, the combination of Eudragit L100-55 and Eudragit L100 overcomes this stickiness effect caused by PG and PEG together (B1–B4). To study the effect of some proposed permeation enhancers on film property, 1% each of limonene, eucalyptol, or diethylene glycol monoethyl ether was added but did not affect the film properties (C2–C3), while the addition of SLS 1% in C1 gave a sticky film. The EM in concentrations (2–5%) was added as a proposed permeation enhancer. A1 was a non-sticky and non-flexible film, but after the addition of the EM in D1, it was still non-sticky but became flexible. C2 formed a non-sticky film, but after the addition of the EM in D4, it became very sticky. In general, the EM can increase film flexibility and stickiness. The results are shown in Table 3.

#### 3.2.2. Successful Trial Formulation STFs

The formulations that met the criteria were chosen and subsequently characterized by pH, viscosity, and elasticity. The results of pH, viscosity, and Young’s modulus for the STFs are shown in Table 4. The pH was performed as a quality control (QC) test. The pH ranges between 3.4 and 5.1. The viscosity ranges between 5.1 and 14.3 cp. These values are relatively small and acceptable. It can be described as a water-like solution if compared with the viscosity of water [31]. The material rigidity was evaluated and the calculated values of E are shown in Table 4. Rigid material had a high E value. A2, B4, and D4 films were the least rigid films, stretched easily under a small load (low stress), and had low E values [41]. Both C4 and D1 showed greater resistance to stretching. A higher load was needed to reach the same stain and they had higher E values. P3, B1, C2, and C3 films were stiffer, they showed much more resistance to stretching, and their E values were much higher, especially the C2 film. P1 and B3 were very soft; even a very small load (low stress) resulted in very large elongation (high strain) that exceeded the elastic region. Although we could not determine E for them by this method, we would expect them to have very low E values. The use of PEG 400, PG, OA, EM, and transcutol reduced the E value, and a further decrease was obtained when used together. However, an increase in the proportion of the polymer or the addition of limonene or eucalyptol increases the value of E.

### 3.3. Vitamin D Loading

According to the findings in Table 5, the addition of vitamin D3 (6000 IU/mL) did not significantly alter the film’s characteristics. Eudragit L100-55 in the B1′ formulation was difficult to dissolve in the presence of 1% of PVP and vitamin D3, so we decreased the concentration of PVP from 1% to 0.5% in the B1′ (new) formulation. All primary films (P3′, B1′ (new), C2′ and D4′) still met the acceptance criteria: the drying time ranged from 3 min and 30 s to 3 min and 45 s, and they were non-sticky, transparent, and had good adhesiveness and flexibility.

### 3.4. In Vitro Release Study Using Franz Diffusion Cell

To investigate and better understand vitamin D3 release from both liquid and dry phases, release tests for both liquid formulations and pre-prepared dry films were conducted. In the liquid formulation, 2 mL of the solution was utilized for this purpose and remained as solution during the experiment; however, for the dry film, the film was prepared before starting the release study using 2 mL solution. The amount released from liquid formulations was much more than that from dry films, according to the data shown in Figure 6 and Figure 7. This can be explained by the low viscosity of the formulation at this stage and the saturation or super-saturation driving force that can be formed with the continuous solvent evaporation [42,43]. The small molecular weight of vitamin D3 is suitable for transdermal delivery (<500 Dalton [44]). However, the polymeric backbone of the formed films Eudragit L100-55 and PVP have high molecular weight (250,000 [45] and 50,000 Dalton [40], respectively). Thus, the release from dry films was a good indication of the continuous ability of the film to free vitamin D3 from the formulation and become available for skin penetration. Additionally, as a result of solvent evaporation, the concentration of vitamin D in the formed membrane is high and thus increases the possibility of drug release.

Figure 6 shows the release from liquid formulations. The amount released in six hours was the highest in the case of the P3′ liquid formulation. The increase in the percentage of Eudragit L100-55 may have counteracted the driving force formed by saturation and masked the effect of the hydrophilic polymer PVP which was expected to promote the release. The same thing was seen in D4′ where the Eudragit percentage was 8%. EM with OA in D4′ and Limonene in C2′ had a negative effect on the amount released; this may be due to the hydrophobic nature of these agents that favors the solubilization of the hydrophobic vitamin D3 in the formulation, thus delaying its release. Figure 7 shows the release from dry films. The amount released from the polymeric film #1′ which contained Eudragit L100-55 5% and vitamin D3 (6000 IU/mL) without any other components was used as a reference. The use of PG alone in the P3′ film had the highest release in unit time, as seen previously in the liquid state. In addition to PG’s solubilizing and plasticizing effects, it also acts as a penetration enhancer [31]. Vitamin D3 is sparingly soluble in PG [16], while it is considered soluble in the receptor fluid ethanol, PBS pH 7.4 (110.22 ± 3.02 μg/mL) [24]; this promotes the release of vitamin D3 and its penetration through the polyamide membrane to reach the receptor fluid where it is more soluble under sink conditions. The release from C2′ and B1′ (new) films was also better than that for the #1′ film. However, in comparison with P3′, the addition of Limonene in C2′ and the increase in the total polymer concentration in the B1′ (new) film reduced the release rate. The combination of the EM and OA together with increasing polymer concentration in the D4′ film had a negative effect on the release.

We could see that the release from formulation C2′ moved to second after P3′ when the film membrane was compared to the solution; this may be attributed to the lower ability of limonene to solubilize vitamin D3 compared to the eutectic mixture and oleic acid combined, with respect to their hydrophobicity and quantity in the formulations.

### 3.5. In Vitro Permeation Study Using Strat-M^®^ Membrane

A 0.5 mL sample was used in the donor compartment to enable studying the permeation from both liquid and dry states over 24 h. Figure 8 shows that the penetration of vitamin D3 from FFS was biphasic. During phase one (0.5–2 h), while the sample was in the liquid state, the evaporating solvent pushed vitamin D3 through the artificial skin membrane, resulting in a faster permeation rate, which is represented by steep slopes. It is expected that vitamin D3 saturation or supersaturation in the film due to the evaporation of the solvents was the driving force for vitamin D3 to leave the formulation and to find an escape, which was the artificial skin. In phase two, a dry film was formed, the penetration slowed down gradually, and the amount permeated through the artificial skin decreased significantly. However, the system reached a steady state again after 5–6 h, where a linear relationship coexisted (R^2^~0.999). The occlusive film formed a drug reservoir that continued with a constant depot flux. Vitamin D3 molecules continued to flow toward the receptor fluid, where sink conditions prevailed until the end of the study (24 h) for all formulations [27,44].

Compared with P3′, the addition of Limonene or PVP in C2′ and B1′ (new), respectively, had no significant effect on the amount permeated (Q). However, the addition of EM and OA at D4′ decreased the release rate. The Q permeated (after 24 h through S = 3.14 cm^2^) from P3′, C2′, and B1′ (new) was significant (about 800 IU under the experimental conditions), while in case of D4′, the Q permeated decreased nearly by half. This can be explained by the ability of EM and OA to dissolve vitamin D3 in relatively significant amounts, which counteracts the main driving force of penetration.

Film-forming solutions contain solvents that could disrupt the intercellular lipids, and while evaporating, they form a driving force through concentrating vitamin D3 in the formulation. This will result in the enhancement of drug permeation by more than one mechanism compared to other formulations.

### 3.6. Stability Studies

All formulations were stored in different conditions in order to evaluate the physical and chemical stability profile of the vitamin D3 and the formulations. All primary formulations were physically stable under the studied storage conditions during the incubation period. They have remained clear, transparent, and without any precipitation. The vitamin D3 assays for all formulations were between 98.0% and 102.0% (<±5% from initial), except D4′, which was unstable under intermediate and under accelerated storage conditions after incubation for 2 and 3 weeks.

## 4. Conclusions

The FFS is a promising delivery system for vitamin D3. It could provide an alternative for oral and parenteral routes. The highly lipophilic vitamin D3 penetrated the artificial skin (Strat-M^®^) over 24 h (about 800 IU across 3.14 cm^2^ under the conditions of the experiment). After the application of the FFS containing the binary solvent ethanol:acetone, it evaporated and the concentration is thought to be increased towards saturation or even super-saturation. The thermodynamic activity increased in the nonvolatile solvent, and this had a great impact and was considered a significant driving force for vitamin D3 release and penetration. Ethanol and acetone also can improve the penetration through the disruption of the intercellular lipid domain. The addition of limonene and PVP in C2′ and B1′ (new), respectively, did not increase the penetration significantly, while the use EM with OA and increasing the hydrophobic polymer concentration had a negative effect on vitamin D3 penetration. Further membrane integrity studies are recommended.

## Figures and Tables

**Figure 1 pharmaceutics-15-00039-f001:**
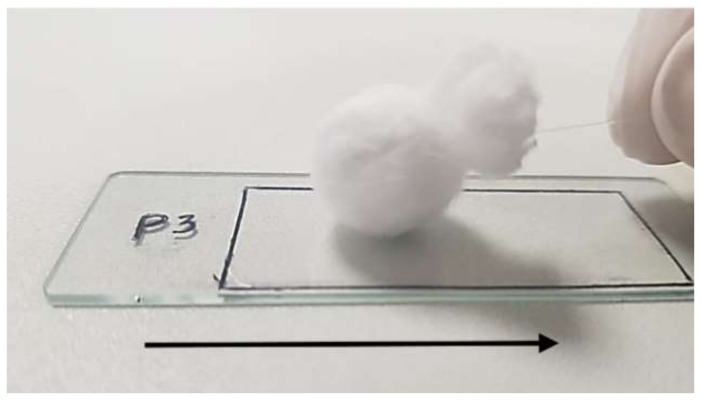
Stickiness evaluation test. The arrow shows the direction of pulling the cotton ball.

**Figure 2 pharmaceutics-15-00039-f002:**
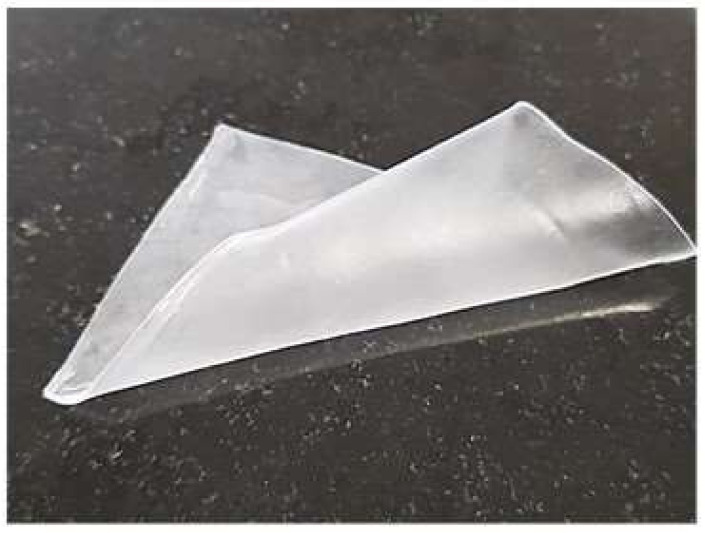
A thick film was rolled up to assess its flexibility.

**Figure 3 pharmaceutics-15-00039-f003:**
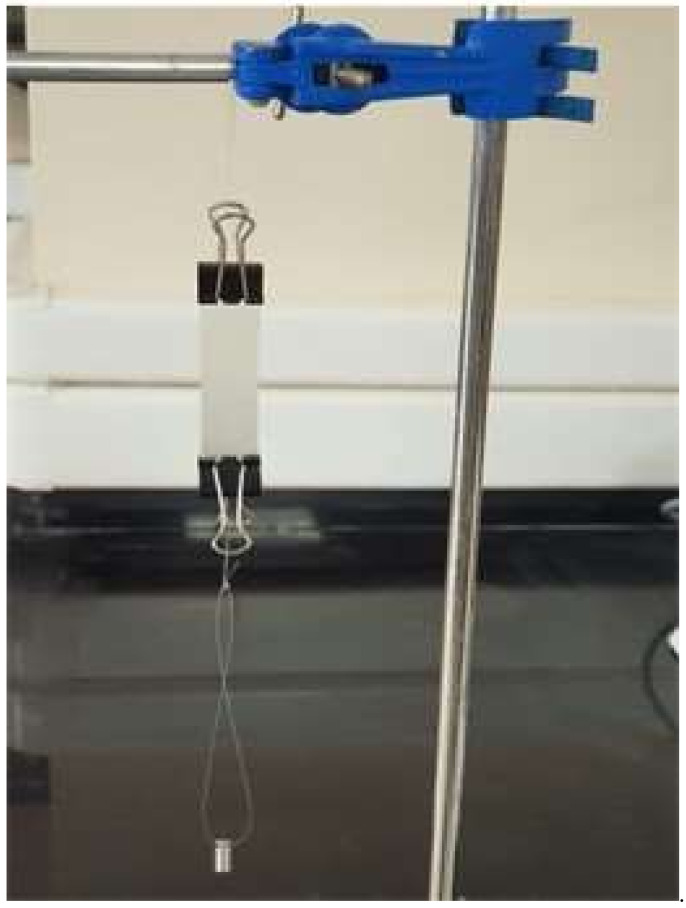
A rectangular polymer hanging from one end on a stand for Young’s modulus determination.

**Figure 4 pharmaceutics-15-00039-f004:**
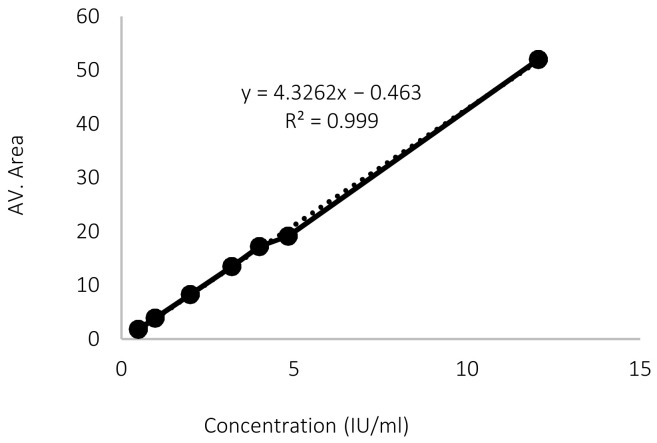
Calibration curve for low vitamin D3 concentrations.

**Figure 5 pharmaceutics-15-00039-f005:**
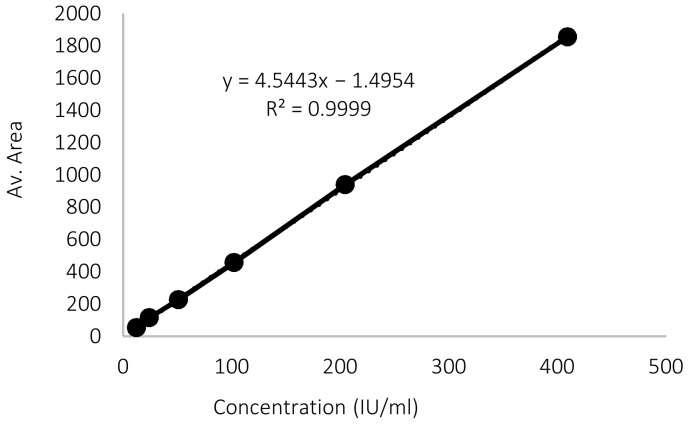
Calibration curve for high vitamin D3 concentrations.

**Figure 6 pharmaceutics-15-00039-f006:**
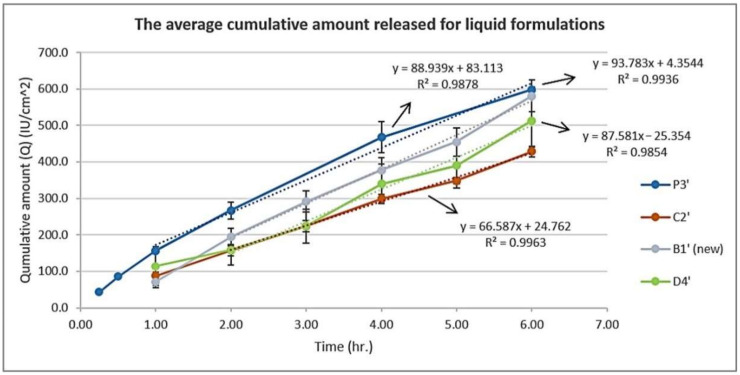
The average cumulative amount released per unit area vs. time for liquid formulations (P3′, C2′, B1′ (new), and D4′).

**Figure 7 pharmaceutics-15-00039-f007:**
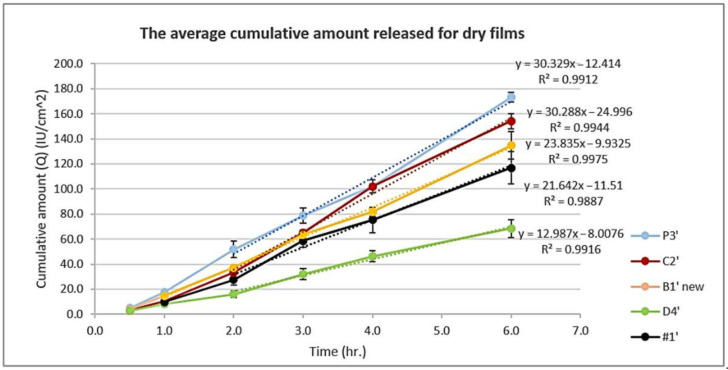
The average cumulative amount released per unit area vs. time for dry films (P3′, C2′, B1′ (new), D4′, and #1).

**Figure 8 pharmaceutics-15-00039-f008:**
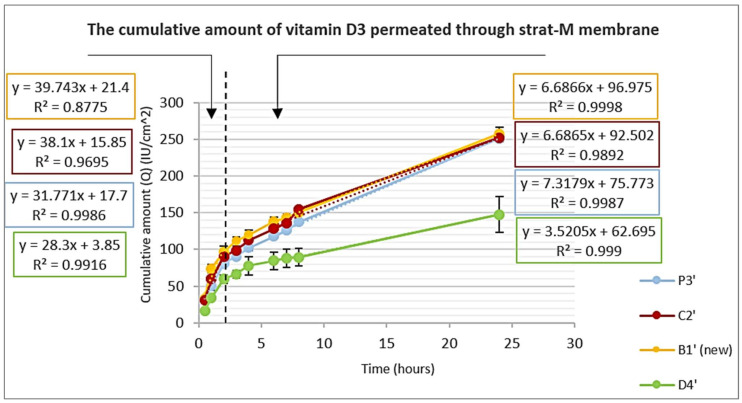
Average cumulative amount permeated per unit area per unit time from P3′, C2′, B1′ (new), and D4′ formulations through Strat-M^®^ membrane.

**Table 1 pharmaceutics-15-00039-t001:** The acceptance criteria for the selection of PTFs after solvent evaporation.

Parameter	Acceptance Criteria
Drying time	Less than 5 min [31]
Stickiness	Not sticky [31,32,33]
Cosmetic appearance	Clear, transparent or semi-transparent [31,32,33,34,35]
Adhesiveness	Adhesive [34,35]
Flexibility	Flexible [31,33,34,35]

**Table 2 pharmaceutics-15-00039-t002:** Primary trial formulations’ composition.

No.	Eudragit			PVP	PEG	PG	OA	IPM	SLS	limonene	Eucalyptol	Transcutol	EM	Ethanol	Ethanol:Acetone (80:20)
	L100-55	L100	S100												
X1	5%	-	-	-	-	-	-	-	-	-	-	-	-	Up to volume	*-*
X2	-	5%	-	-	-	-	-	-	-	-	-	-	-		*-*
X3	-	-	5%	-	-	-	-	-	-	-	-	-	-		*-*
X4	-	-	-	5%	-	-	-	-	-	-	-	-	-		*-*
XP1	5%	-	-	5%	2%	-	-	-	-	-	-	-	-		*-*
XP2	5%	-	-	5%	-	-	2%	-	-	-	-	-	-		*-*
XP3	5%	-	-	5%	-	2%	-	-	-	-	-	-	-		*-*
XP4	-	5%	-	-	2%	-	-	-	-	-	-	-	-		*-*
XP5	-	5%	-	-	-	2%	-	-	-	-	-	-	-		*-*
XP6	-	5%	-	-	-	-	2%	-	-	-	-	-	-		*-*
XP7	-	-	5%	-	2%	-	-	-	-	-	-	-	-		*-*
XP8	-	-	5%	-	-	2%	-	-	-	-	-	-	-		*-*
XP9	-	-	5%	-	-	-	2%	-	-	-	-	-	-		*-*
1	5%	-	-	-	-	-	-	-	-	-	-	-	-	*-*	Up to volume
2	-	5%	-	-	-	-	-	-	-	-	-	-	-	*-*	
3	-	-	5%	-	-	-	-	-	-	-	-	-	-	*-*	
4	-	-	-	5%	-	-	-	-	-	-	-	-	-	*-*	
P1	5%	-	-	-	2%	-	-	-	-	-	-	-	-	*-*	
P2	5%	-	-	-	-	-	2%	-	-	-	-	-	-	*-*	
P3	5%	-	-	-	-	2%	-	-	-	-	-	-	-	*-*	
P4	-	5%	-	-	2%	-	-	-	-	-	-	-	-	*-*	
P5	-	5%	-	-	-	2%	-	-	-	-	-	-	-	*-*	
P6	-	5%	-	-	-	-	2%	-	-	-	-	-	-	*-*	
P7	-	-	5%	-	2%	-	-	-	-	-	-	-	-	*-*	
P8	-	-	5%	-	-	2%	-	-	-	-	-	-	-	*-*	
P9	-	-	5%	-	-	-	2%	-	-	-	-	-	-	*-*	
A1	8%	-	-	-	-	2%	1%	-	-	-	-	-	-	*-*	
A2	8%	-	-	-	2%	-	1%	-	-	-	-	-	-	*-*	
A3	8%	-	-	-	2%	2%	-	-	-	-	-	-	-	*-*	
A4	8%	-	-	-	-	2%	-	2%	-	-	-	-	-	*-*	
A5	8%	-	-	-	2%	-	-	2%	-	-	-	-	-	*-*	
A6	-	8%	-	-	2%	-	-	2%	-	-	-	-	-	*-*	
A7	-	5%	-	-	2%	-	1%	-	-	-	-	-	-	*-*	
B1	6%	-	-	1%	-	2%	-	-	-	-	-	-	-	*-*	
B1 new	6%	-	-	0.5%	-	2%	-	-	-	-	-	-	-	*-*	
B2	4%	4%	-	-	2%	2%	-	-	-	-	-	-	-	*-*	
B3	6%	2%	-	-	2%	2%	-	-	-	-	-	-	-	*-*	
B4	7%	1%	-	-	2%	2%	-	-	-	-	-	-	-	*-*	
C1	5%	-	-	-	-	2%	-	-	1%	-	-	-	-	*-*	
C2	5%	-	-	-	-	2%	-	-	-	1%	-	-	-	*-*	
C3	5%	-	-	-	-	2%	-	-	-	-	1%	-	-	*-*	
C4	5%	-	-	-	-	2%	-	-	-	-	-	1%	-	*-*	
D1	8%	-	-	-	2%	-	1%	-	-	-	-	-	2%	*-*	
D2	8%	-	-	-	2%	-	1%	-	-	-	-	-	5%	*-*	
D3	5%	-	-	-	-	2%	-	-	-	1%	-	-	5%	*-*	
D4	8%	-	-	-	-	2%	1%	-	-	-	-	-	5%	*-*	
D5	6%	-	-	1%	-	2%	-	-	-	-	-	-	5%	*-*	

PVP: polyvinylpyrrolidone, PEG: polyethylene glycol 400, PG: propylene glycol, OA: oleic acid, IPM: isopropyl myristate, SLS: sodium lauryl sulfate, EM: eutectic mixture of camphor and menthol (1:1).

**Table 3 pharmaceutics-15-00039-t003:** Primary trial formulations’ characterization and evaluation.

No.	Drying Time	Stickiness	Cosmetic Appearance (Thin Film)	Adhesiveness	Flexibility	Pass/Fail
X1	5 min 15 s	Non-sticky	Transparent, clear	Low	Not flexible	Fail
X2	4 min 30 s	Not sticky	White	Low	Not flexible	Fail
X3	6 min 45 s	Not sticky	White	Low	Not flexible	Fail
X4	4 min	Not sticky	Transparent, clear	High	N.A.	Fail
XP1	5 min 45 s	Non-sticky	Transparent	High	High	Fail
XP2	5 min 45 s	Non-sticky	Transparent	Low	Not flexible	Fail
XP3	6 min	Non-sticky	Transparent	Good	Good	Fail
XP4	5 min 45 s	Non-sticky	Transparent	Low	Not flexible	Fail
XP5	5 min 45 s	Non-sticky	Transparent	Low	Not flexible	Fail
XP6	5 min 30 s	Non-sticky	Transparent	Low	Not flexible	Fail
XP7	5 min	Non-sticky	Transparent	Low	Low	Fail
XP8	5 min 15 s	Non-sticky	Transparent	Low	Not flexible	Fail
XP9	5 min	Non-sticky	Transparent	Low	Low	Fail
P1	4 min and 30 s	Non-sticky	Transparent	High	High	Pass
P2	4 min and 30 s	Non-sticky	Transparent	Low	Not flexible	Fail
P3	4 min and 30 s	Non-sticky	Transparent	Good	Good	Pass
P4	3 min and 45 s	Non-sticky	Transparent	Low	Not flexible	Fail
P5	3 min and 45 s	Non-sticky	Transparent	Low	Not flexible	Fail
P6	3 min and 30 s	Non-sticky	Transparent	Low	Not flexible	Fail
P7	3 min	Non-sticky	Transparent	Low	Low	Fail
P8	3 min	Non-sticky	Transparent	Low	Not flexible	Fail
P9	3 min	Non-sticky	Transparent	Low	Low	Fail
A1	5 min 30 s	Non-sticky	Transparent	Good	Not flexible	Fail
A2	3 min 30 s	Non-sticky	Transparent	Good	High	Pass
A3	5 min	Sticky	Transparent	High	High	Fail
A4	4 min 14 s	Non-sticky	Transparent	Good	Low	Fail
A5	4 min	Non-sticky	Transparent	Good	Good	Fail
A6	2 min 45 s	Not sticky	Transparent	Low	Good	Fail
A7	3 min 30 s	Not sticky	Transparent	Low	Low	Fail
B1	3 min 15 s	Not sticky	Transparent,	Good	Good	Pass
B1 new	3 min 15 s	Not sticky	Transparent,	Good	Good	Pass
B2	4 min 15 s	Not sticky	Transparent, not smooth	Good	High	Fail
B3	3 min 15 s	Not sticky	Transparent	Good	Good	Pass
B4	3 min	Not sticky	Transparent	High	High	Pass
C1	3 min 15 s	Sticky	Semi-Transparent	Good	Good	Fail
C2	3 min 30 s	Not sticky	Transparent	Good	Good	Pass
C3	3 min 30 s	Not sticky	Transparent	Good	Good	Pass
C4	4 min	Not sticky	Transparent	Good	Good	Pass
D1	3 min 30 s	Non-sticky	Transparent	Good	Good	Pass
D2	4 min 30 s	Slightly sticky	Transparent	Good	Good	Fail
D3	3 min 45 s	Very sticky	Transparent	Good	Good	Fail
D4	3 min 45 s	Non-sticky	Transparent	Good	Good	Pass
D5	4 min 15 s	Slightly sticky	Transparent	Good	Good	Fail

**Table 4 pharmaceutics-15-00039-t004:** The results of pH, viscosity, and Young’s modulus for STFs.

No.	Ingredients Dissolved in Ethanol and Acetone (80:20)	pH	Viscosity (CP)	Young’s Modulus kPa
P1	Eudragit L100-55 5% + PEG 2%	3.5 ± 0.1	5.1 ± 0.0	N.A.
P3	Eudragit L100-55 5% + PG 2%	3.9 ± 0.1	5.5 ± 0.0	1114.6
A2	Eudragit L100-55 8% + PEG 2% + OA 1%	3.6 ± 0.1	8.2 ± 0.1	87.4
B1	Eudragit L100-55 6% + PVP 1% + PG 2%	3.5 ± 0.1	8.7 ± 0.0	1459.0
B3	Eudragit L100 2% + Eudragit L100-55 6% + PG 2% + PEG 2%	3.4 ± 0.1	12.1 ± 0.1	N.A.
B4	Eudragit L100 1% + Eudragit L100-55 7% + PG 2% + PEG 2%	3.4 ± 0.1	18.9 ± 0.0	82.9
C2	Eudragit L100-55 5% + PG 2% + limonene 1%	4.1 ± 0.1	5.5 ± 0.0	12,598.0
C3	Eudragit L100-55 5% + PG 2% + Eucalyptol 1%	5.0 ± 0.1	5.6 ± 0.0	3785.7
C4	Eudragit L100-55 5% + PG 2% + transcutol 1%	4.2 ± 0.1	5.8 ± 0.0	705.3
D1	Eudragit L100-55 8% + PEG 2% + OA 1% + EM 2%	4.7 ± 0.1	14.3 ± 0.1	264.0
D4	Eudragit L100-55 8% + PG 2% + OA 1% + EM 5%	5.1 ± 0.1	11.1 ± 0.2	55.5

**Table 5 pharmaceutics-15-00039-t005:** Characterization results of primary formulations containing vitamin D3 6000 IU/mL, using ethanol:acetone (80:20) as a solvent.

	Ingredients	Drying Time	Stickiness	Cosmetic Appearance	Adhesiveness	Flexibility
P3′	Eudragit L100-55 5% + PG 2% + vitamin D3 6000 IU/mL	3 min 30 s	Non-sticky	Transparent	Good	Good
B1′ (new)	Eudragit L100-55 6% + PVP 0.5% + PG 2% + vitamin D3 6000 IU/mL	3 min 45 s	Non-sticky	Transparent	Good	Good
C2′	Eudragit L100-55 5% + PG 2% + limonene 1% + vitamin D3 6000 IU/mL	3 min 30 s	Non-sticky	Transparent	Good	Good
D4′	Eudragit L100-55 8% + PG 2% + OA 1% + EM 5% + vitamin D3 6000 IU/mL	3 min 45 s	Non-sticky	Transparent	Good	Good

## Data Availability

Not applicable.

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
