# Peer review of "Preparation and Evaluation of Vitamin D3 Supplementation as Transdermal Film-Forming Solution"

_pharmaceutics, 2022, doi:10.3390/pharmaceutics15010039_

Round 1
Reviewer 1 Report
In this study entitled “Preparation and evaluation of vitamin D3 supplementation as transdermal film forming solution” the authors tried to optimize a film with topical application to release Vitamin D3.
This is an interesting manuscript. However, some points have to be improved:
1) The word percutaneous is as a keyword, as well as in the introduction the skin and the importance of the stratum corneum are described. Also, the word penetration has been used in the paper.
However, the research does not use any skin, only artificial membrane.
This part of the introduction should be modified.
2) Missing purchase of Eudragits, PEG, PG, SLS, camphor and menthol (lines 108-110)
3) All relevant analytical parameters of HPLC method are missing: LDD, LDQ, calibration curve, retention time, etc.. (section 2.3).
4) The way of exposing the equations must be consistent with each other.
There are only two equations, so there is an error on line 229 with the equation number.
5) How is Vitamin D loaded in the films? Section 2.8 needs to be expanded.
6) With regard to in vitro permeation studies:
a. Receptor fluid is usually at 37°C. Why do you use it at 32°C?
b. What is the volume of the acceptor compartment? What is the application area? It must be indicated in section 2.9.
c. Also indicate the sampling times in the case of polyamide membranes. What is the supplier of these membranes?
d. In the results a reference membrane is mentioned, which is not described in the methodology.
e. How has the cumulative amount of the release been calculated?
f. Why is a different volume of formulations applied in in vitro release studies using polyamide and StratM membranes? If the test is carried out with an infinite dose, it is better to apply the same amount to be able to compare results.
g. Why the sampling times were different in both release tests?
h. Has any statistical treatment been applied to the results of the release test to support the conclusions?
7) Stability studies must be more detailed: temperatures, measurement time, parameters to be measured, how they have been accelerated, if they follow any regulations, etc. In the results section, at least one table must be shown with the initial and final values ​​to be able to compare them.
8) The pH values ​​of the formulations are between 3.4 and 5.1, are they lower than the value of the skin surface, 5.5, could they irritate the skin? Has it been verified?
9) The header of the tables should provide more information.
10) In the results appears a formulation B1' new, which there is no description. Why is there a new B1 formulation?
11) The results should be compared, if possible, with more bibliographical references.
12) Errors in the references:
a. Reference 30 in line 234, is missing
b. Reference 31 in line 245 is wrong
c. From reference 30, all the bibliography must be corrected
d. Reference 37 is missing
13) Format of some words in the article must be revised:
Vitamin D3 in line 23 (missing the letter V), some capitals letters in lines 48, 148, 176, 202,…
The document lacks consistency in terms of format of titles/ tables/ figures
Reviewer 2 Report
The paper evaluated transdermal film formulations for the delivery of vitamin D3. Experiments were performed to characterize the drying time, stickiness, appearance, adhesiveness, flexibility, pH, viscosity, elasticity, and vitamin D3 loading of the formulations. Vitamin D3 release from the film and formulations and its permeation across Strat M membrane were also determined. The “promising” delivery formulations were identified. In general, there is a lack of discussion of the results in the paper. In addition, some methods used to characterize the formulations are not conventional that they are not quantitative to meet the industrial standards. The use of Strat M membrane in the study to conclude effective transdermal delivery will require justification. There are also typos and grammatical errors in the paper. The following are the point-by-point comments.
1. There are too much background information in the Introduction. For example, the details on the skin structure and properties are not necessary.
2. The drying time study was performed with 0.1 mL formulations on 10 cm2 (10 uL per cm2) but the vitamin D3 release and Strat M permeation studies were performed with 2 mL formulations. Can the results from the drying time study be used to interpret the data in the vitamin D3 release and Strat M permeation studies? This inconsistency should be discussed.
3. The approaches in the stickiness, adhesiveness, and flexibility studies are not conventional. They are not quantitative. What are the references and controls in these studies? What are the meanings of “low,” “good,” and “high”? Without the definitions and quantitation, it is difficult for the readers to understand the results.
4. There is no discussion of the results in the paper. In its current form, the paper merely presents the data, which can be categorized as screening.
5. The use of Ostwald viscometer to measure the viscosity of the formulations may not be adequate. The shear stress vs. shear rate relationship is missing and such relationship can be important in skin formulation characterization.
6. There are too many picture figures that may not be necessary (this can be related to the unconventional experimental approaches). The text has already described the experimental setup.
7. Please provide the dimensions of the Franz diffusion cell. For example, what is the diameter of the opening? What is the volume of the receptor chamber? Is the ethanol:PBS (50:50) solution based on w/w or v/v?
8. The use of 50% ethanol in the receptor fluid would likely affect the barrier of the Strat M and affect vitamin D3 permeation. This is not representative of the physiological condition. This reviewer understands the need to increase the solubility of vitamin D3 in the receptor chamber, but the permeation enhancement effect needs to be considered.
9. This reviewer does not agree with the interpretation of the permeation study and the conclusion that enough vitamin D3 can be delivered across the skin using the proposed formulations in practice. First, it is well known that Strat M can be “leakier” than skin. Second, the use of 50% ethanol in the receptor chamber can enhance vitamin D3 permeation. The similar amounts of vitamin D3 delivered in the release study vs. the Strat M study (less than 3x difference) can be a result of this.
10. Data are needed to support the statement on “saturation or even supersaturation” in the Conclusion.
11. Some formats need to be corrected (e.g., bullets in Section 3.1.1). In addition, there are too many tables, and they should be combined.
Reviewer 3 Report
Transdermal film forming solution FFS - is a new form of the drug intended for transdermal administration of an active substance to the skin. This form can be sprayed or applied to the skin, and upon evaporation of the vehicle, a film is formed on the skin which acts as a matrix for sustained release of drug to the skin.
The aim of this study was to formulate FFS with vitamin D3 and to assess the physicochemical properties and study the release and permeation through the skin in vitro. The study is interesting, but I have some comments:
1. Funding should be completed.
2. Page 2, line 69 - no reference in the literature, “…..G. Costa et al used several ………”
3. Page 2, line 87 - no reference in the literature: “……Other researchers used the polymeric nanoparticle (TyroSphere) and coated……………………..”
4. Page 2, line 110; ” …….all pharmaceutical grade”?. Please specify which reagent of which purity class.
5. Data on manufacturers of reagents and apparatus should be completed.
6. In chapter 2.5 “General method of preparation”. Complete the sonification process parameters.
7. Has the slide been viewed under a microscope? Has the particle size of the active substance been measured?
8. Why a mixture of ethanol and BPS (50:50) was chosen as the acceptor fluid for the release studies?
9. Why was the release study conducted for both the liquid formulations and for the dry films?
10. Please describe in more detail the conditions under which the stability study was conducted.
11. What parameters or criteria were used to evaluate the stability of the preparations?
12. Literature references should be written in accordance with the journal's requirements: (Author 1, A.B.; Author 2, C.D. Title of the article. Abbreviated Journal Name Year, Volume, page range.).
13. A skin irritation test would have to be performed for the formulations prepared.
14. No citation of references: 24, 25, 26, 27, 28.
15. There is no reference 37 in the literature list.
16. The manuscript was not prepared with care in terms of editing.
Round 2
Reviewer 2 Report
Most responses to the reviewers’ comments are satisfactory. However, some of these responses are not included in the text of the paper. In addition, the following concerns need to be addressed.
#2, The explanation by the authors is not convincing. If the conditions are different (0.1 mL vs. 0.5 mL or 2 mL), the amount of vitamin D3 delivered will likely to be different. The data from using 0.5 mL cannot be used to assess vitamin D3 delivery in practice. Additional release and permeation experiments with 10 uL per cm2 dosing are needed. Or, possible different outcomes from 10 uL per cm2 dose (vs. the results in Figs. 6 and 7 using 2 mL and 0.5 mL doses) should be discussed.
#8, The use of 50% ethanol in the receptor fluid can be considered as a “flaw” in the study unless it can be shown that 50% ethanol does not affect the skin barrier (ethanol as a penetration enhancer). Just because some studies used the same condition does not mean that it is correct and that the problem can be ignored. Additional experiment to show that the receptor solution does not affect skin permeability is needed. Or, this problem should be noted and discussed.
#10, It is correct that film forming solution can create a saturated condition in the donor chamber, but the vitamin D3 release study used 2 mL and permeation study used 0.5 mL solution. It is unclear whether the 0.5 and 2 mL solutions created a film in those studies to create a saturated condition.
Figures of HPLC calibration curves (unless they are abnormal) are not normally presented in scientific papers.
Author Response
Dear reviewer:
We appreciate your input. We carefully evaluated the valuable comments and adjusted the manuscript as possible. Our point-by-point responses and the related adjustments are attached. Any further comments will be addressed.
Please be aware that the Changes Outline report addresses the page and line numbers when the manuscript's track changes feature is turned off
Please see attached file
Thank you

Reviewer 3 Report
-
- - Line 527: “ ……….The assay % for all formulations were (< ±5% ……………….” There is no information about what was the subject of the test.
- - What is the contribution of new co-author Ramzi Muqedi to the article?
Author Response
Dear Editor/ reviewer
We appreciate the reviewers and editor input. We carefully evaluated the valuable comments and adjusted the manuscript as possible.
Our point-by-point responses and the related adjustments are attached. Any further comments will be addressed.
Please be aware that the Changes Outline report addresses the page and line numbers when the manuscript's track changes feature is turned off
Please see attached file
Thank you

Round 3
Reviewer 2 Report
The authors seem to misunderstand the reviewer’s concerns. The strategy in the formulation development is understandable in the paper. However, the different conditions in the formulation development and those in transdermal delivery in practice should be highlighted. For the concern on dosing volume, the different dosing volumes in the in vitro permeation study (0.5 mL on 3.14 cm2 or 159 uL/cm2) and that proposed in practice (100 uL on 10 cm2 or 10 uL/cm2, used in the evaporation study) are important because such differences (in dosing volume) can affect the thermodynamic activity of the drug in the formulation (how fast the solvent was evaporated) in the in vitro permeation study using Strat-M®. Or, did the authors intend to use 0.5 mL dose in practice? If a large volume is used, this will require long evaporation time to form the transdermal film. The conclusion in the paper should be supported by the data. For example, the conclusion (Line 464-469) implies that the solvent evaporated in 5 min after application in the permeation study and provided “acceptable amount over 24 hours” (800 IU across 3.14 cm2), which is misleading. Considering the impact of evaporation, the descriptions in the Conclusion need to be more specific. There are also other factors such as the use of ethanol in the receptor and Strat-M instead of actual skin that can affect the “800 IU across 3.14 cm2” delivery. This can be revised by stating the specific conditions or using phrases such as “under the conditions in this study” in the text.
Finally, a check on English use is required.
Author Response
Dear Reviewer:
We appreciate your input. We carefully evaluated the valuable comments and adjusted the manuscript as possible. Our point-by-point responses and the related adjustments are attached. Any further comments will be addressed.
English language was reviewed
Please be aware that the Changes Outline report addresses the page and line numbers when the manuscript's track changes feature is turned On.
Thank You
Dr Moammal Qurt
